# MANDERA: Malicious Node Detection in Federated Learning via Ranking

## Abstract

Federated Learning is a distributed learning paradigm which seeks to preserve the privacy of each participating node's data. However, federated learning is vulnerable to attacks, specifically to our interest, model integrity attacks. In this paper, we propose a novel method for malicious node detection called MANDERA. By transferring the original message matrix into a ranking matrix whose column shows the relative rankings of all local nodes along different parameter dimensions, our approach seeks to distinguish the malicious nodes from the benign ones with high efficiency based on key characteristics of the rank domain. We have proved, under mild conditions, that MANDERA is guaranteed to detect all malicious nodes under typical Byzantine attacks with no prior knowledge or history about the participating nodes. The effectiveness of the proposed approach is further confirmed by experiments on three classic datasets, CIFAR-10, FASHION-MNIST and MNIST. Compared to the state-of-art methods in the literature for defending Byzantine attacks, MANDERA is unique in its way to identify the malicious nodes by ranking and its robustness to effectively defense a wide range of attacks.

## 1 Introduction

Federated learning (FL) has observed a steady rise in use across a plethora of applications. FL departs from conventional centralized learning by allowing multiple participating nodes to learn on a local collection of training data, before each respective node's updates are sent to a global coordinator for aggregation. The global model collectively learns from each of these individual nodes before relaying the updated global update back to the participating nodes. With an aggregation of multiple nodes, the resulting model observes greater performance than if each node was to learn on their local subset only. FL presents two key advantages, increased privacy for the contributing node as local data is not communicated to the global coordinator, and a reduction in computation by the global node as the computation is offloaded to contributing nodes.

However, the presence of malicious actors in the collaborative process may seek to poison the performance of the global model, to reduce the output performance of the model (Chen et al., 2017; Fang et al., 2020; Tolpegin et al., 2020b), or to embed hidden back-doors within the model (Bagdasaryan et al., 2020). Byzantine attack aims to devastate the performance of the global model by manipulating the gradient values of malicious nodes in a certain fashion. As these attacks emerged, researchers seek to defend FL from the negative impacts of these attacks.

In the literature, there are two typical defense strategies: malicious node detection and robust learning. Malicious node detection defenses by detecting malicious nodes and removing them from the aggregation (Blanchard et al., 2017; Guerraoui et al., 2018; Li et al., 2020; So et al., 2021). Robust learning (Blanchard et al., 2017; Yin et al., 2018; Guerraoui et al., 2018; Fang et al., 2020; Cao et al., 2020), however, withstands a proportion of malicious nodes and defenses by reducing the negative impacts of the malicious nodes via various robust learning methods (Wu et al., 2020b; Xie et al., 2019; 2020; Cao et al., 2021).

In this paper, we focus on defensing Byzantine attacks via malicious node detection. In the literature, there have been a collection of efforts along this research line. Blanchard et al. (2017) propose a defense referred to as Krum that treats local nodes whose update vector is too far away from the aggregated barycenter as malicious nodes and precludes them from the downstream aggregation. Guerraoui et al. (2018) propose Bulyan, a process that performs aggregation on subsets of node

updates (by iteratively leaving each node out) to find a set of nodes with the most aligned updates given an aggregation rule. Xie et al. (2019) compute a *Stochastic Descendant Score* (SDS) based on the estimated descendant of the loss function, and the magnitude of the update submitted to the global node, and only include a predefined number of nodes with the highest SDS in the aggregation. On the other hand, Chen et al. (2021) propose a zero-knowledge approach to detect and remove malicious nodes by solving a weighted clustering problem. The resulting clusters update the model individually and accuracy against a validation set are checked. All nodes in a cluster with significant negative accuracy impact are rejected and removed from the aggregation step.

Although the aforementioned methods try to detect malicious nodes in different ways, they all share a common nature: the detection is based on the gradient updates directly. However, it is usually the case that different dimensions of the gradients remain quite different in the range of values and follow very different distributions. This phenomena makes it very challenging to precisely detect malicious nodes directly based on the node updates, as a few dimensions often dominate the final result. Although the weighted clustering method proposed by Chen et al. (2021) could avoid this problem partially by re-weighting different update dimensions, it is often not trivial to determine the weights in a principled way.

In this paper, we propose to resolve this critical problem from a novel perspective. Instead of working on the node updates directly, we propose to extract information about malicious nodes indirectly by transforming the node updates from numeric gradient values to the rank domain. Compared to the original numeric gradient values, whose distribution is difficult to model, the ranks are much easier to handle both theoretically and practically. Moreover, as ranks are scale-free, we no longer need to worry about the scale difference across different dimensions. We proved under mild conditions that the first two moments of the transformed rank vectors carry key information to detect the malicious nodes under a wide range of Byzantine attacks. Based on these theoretical results, a highly efficient method called MANDERA is proposed to separate the malicious nodes from the benign ones by clustering all local nodes into two groups based on the moments of their rank vectors. With the assumption that malicious nodes are the minority in the node pool, we can simply treat all nodes in the smaller cluster as malicious nodes and remove them from the aggregation.

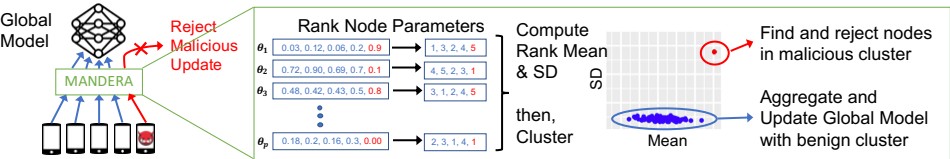

Figure 1: An Overview of MANDERA

The contributions of this work are as follows. **(1)** We propose the first algorithm leveraging the rank domain of model updates to detect malicious nodes (Figure 1). **(2)** We provide theoretical guarantee for the detection of malicious nodes based on the rank domain under Byzantine attacks. **(3)** Our method does not assume knowledge on the number of malicious nodes, which is required in the learning process of prior methods. **(4)** We experimentally demonstrate the effectiveness and robustness of our defense on Byzantine attacks, including Gaussian attack, Sign Flipping attack and Zero Gradient attack, in addition to a more subtle Label Flipping data poisoning attack. **(5)** An experimental comparison between MANDERA and a collection of robust aggregation techniques are provided. The computation times are also compared, demonstrating gains of MANDERA by operating in the rank domain.

## 2 DEFENSE FORMALIZATION

### 2.1 NOTATIONS

Suppose there are $n$ local nodes in the federated learning framework, where $n_1$ nodes are benign nodes whose indices are denoted by $\mathcal{I}_b$ and the other $n_0 = n - n_1$ nodes are malicious nodes whose indices are denoted by $\mathcal{I}_m$. The training model is denoted by $f(\boldsymbol{\theta}, \boldsymbol{D})$, where $\boldsymbol{\theta} \in \mathbb{R}^{p \times 1}$ is a $p$-dimensional parameter vector and $\boldsymbol{D}$ is a data matrix. Denote the message matrix received from all local nodes by the central server as $\boldsymbol{M} \in \mathbb{R}^{n \times p}$, where $\boldsymbol{M}_{i,:}$ denotes the message received from node $i$. For a benign node $i$, let $\boldsymbol{D}_i$ be the data matrix on it with $N_i$ as the sample size, we have $\boldsymbol{M}_{i,:} = \frac{\partial f(\boldsymbol{\theta}, \boldsymbol{D}_i)}{\partial \boldsymbol{\theta}}$. A malicious node $j \in \mathcal{I}_m$, however, tends to attack the learning system by

manipulating $M_{j,:}$ in some way. Hereinafter, we denote $N^* = \min(\{N_i\}_{i \in \mathcal{I}_b})$ to be the minimal sample size of the benign nodes.

Given a vector of real numbers $a \in \mathbb{R}^{p \times 1}$, define its ranking vector as $b = Rank(a) \in perm\{1, \cdots, p\}$, where the ranking operator $Rank$ maps the vector $a$ to its permutation space $perm\{1, \cdots, p\}$ which is the set of all the permutations of $\{1, \cdots, p\}$. For example, $Rank(1.1, -2, 3.2) = (2, 3, 1)$. We adopt average ranking, when there are ties. With the *Rank* operator, we can transfer the message matrix $M$ to a ranking matrix $R$ by replacing its column $M_{:,j}$ by the corresponding ranking vector $R_{:,j} = Rank(M_{:,j})$. Further define

$$e_i \triangleq \frac{1}{p} \sum_{j=1}^{p} R_{i,j} \qquad \text{and} \qquad v_i \triangleq \frac{1}{p} \sum_{j=1}^{p} (R_{i,j} - e_i)^2$$

to be the mean and variance of $R_{i,:}$, respectively. As it is shown in later subsections, we can judge whether node $i$ is a malicious node based on $(e_i, v_i)$ under various attack types. In the following, we will highlight the behaviour of the benign nodes first, and then discuss the behaviour of malicious nodes and their interactions with the benign nodes under various Byzantine attacks respectively.

## 2.2 BEHAVIOUR OF BENIGN NODES

As the behaviour of benign nodes does not depend on the type of Byzantine attack, we can study the statistical properties of $(e_i, v_i)$ for a benign node $i \in \mathcal{I}_b$ before the specification of a concrete attack type. For any benign node $i$, the message generated for $j^{th}$ parameter is

$$M_{i,j} = \frac{1}{N_i} \sum_{l=1}^{N_i} \frac{\partial f(\boldsymbol{\theta}, \boldsymbol{D}_{i,l})}{\partial \boldsymbol{\theta}_j}, \tag{1}$$

where $\boldsymbol{D}_{i,l}$ denotes the $l^{th}$ sample on it. Throughout this paper, we always assume that $\boldsymbol{D}_{i,l}$s are independent and identically distributed (IID) samples drawn from a data distribution $\mathbb{D}$. Under the independent data assumption, since Equation 1 tells us that $M_{i,j}$ is the sample mean of IID random variables, i.e., $\{\frac{\partial f(\boldsymbol{\theta}, \boldsymbol{D}_{i,l})}{\partial \boldsymbol{\theta}_j}\}_{l=1}^{N_i}$, directly applying the Strong Law of Large Numbers (SLLN) and Central Limit Theorem (CLT) leads to the lemma below immediately.

**Lemma 1.** *Under the independent data assumption, further denote $\mu_j = \mathbb{E}(\frac{\partial f(\boldsymbol{\theta}, \boldsymbol{D}_{i,l})}{\partial \boldsymbol{\theta}_j})$ and $\sigma_j^2 = \mathrm{Var}(\frac{\partial f(\boldsymbol{\theta}, \boldsymbol{D}_{i,l})}{\partial \boldsymbol{\theta}_j}) < \infty$, with $N_i$ going to infinity we have for $\forall \ j \in \{1, \cdots, p\}$*

$$M_{i,j} \to \mu_j \ a.s. \quad and \quad M_{i,j} \xrightarrow{d} \mathcal{N}\left(\mu_j, \sigma_j^2/N_i\right). \tag{2}$$

## 2.3 BEHAVIOUR OF MALICIOUS NODE UNDER THE GAUSSIAN ATTACK

**Definition 1** (Gaussian attack). *In a Gaussian attack, the attacker manipulates malicious nodes to send Gaussian random messages to the global coordinator, i.e., $\{M_{i,:}\}_{i \in \mathcal{I}_m}$ are independent random samples from Gaussian distribution $\mathcal{MVN}(\boldsymbol{m}_{b,:}, \Sigma)$, where $\boldsymbol{m}_{b,:} = \frac{1}{n_1} \sum_{i \in \mathcal{I}_b} M_{i,:}$ and $\Sigma$ is the covariance matrix determined by the attacker.*

Considering that $M_{i,j} \to \mu_j$ almost surely (a.s.) with $N_i$ going to infinity for all $i \in \mathcal{I}_b$ based on Lemma 1, it is straightforward to see that $\lim_{N^* \to \infty} \boldsymbol{m}_{b,j} = \mu_j$ a.s., and the distribution of $M_{i,j}$ for each $i \in \mathcal{I}_m$ converges to the Gaussian distribution centered at $\mu_j$. Lemma 2 provides the details.

**Lemma 2.** *Under the same assumption as in Lemma 1, with $N^*$ going to infinity, we have for each malicious node $i \in \mathcal{I}_m$ under the Gaussian attack that*

$$M_{i,j} \xrightarrow{d} \mathcal{N}\left(\mu_j, \Sigma_{j,j}\right), \ 1 \leq j \leq p. \tag{3}$$

Lemma 1 and Lemma 2 tell us that for each parameter dimension $j$, $\{M_{i,j}\}_{i=1}^{n}$ are independent Gaussian random variables with the same mean (i.e, $\mu_j$) but different variances (i.e., $\sigma_j^2/N_i$ or $\Sigma_{j,j}$) under the Gaussian attack. Due to the symmetry of Gaussian distribution, it is straightforward to see

$$\mathbb{E}(R_{i,j}) = \frac{n+1}{2}, \ 1 \leq i \leq n, \ 1 \leq j \leq p.$$

Moreover, the exchangeability of benign nodes and the exchangeability of malicious nodes when $N^*$ is reasonably large tell us: for each parameter dimension $j$, there exist two positive constants $s_{b,j}^2$ and $s_{m,j}^2$ such that

$$\text{Var}(\boldsymbol{R}_{i,j}) = s_{b,j}^2, \ \forall \, i \in \mathcal{I}_b, \quad \text{and} \quad \text{Var}(\boldsymbol{R}_{i,j}) = s_{m,j}^2, \ \forall \, i \in \mathcal{I}_m,$$

where both $s_{b,j}^2$ and $s_{m,j}^2$ are complex functions of $\sigma_j^2$, $\Sigma_{j,j}$ and $\{N_i\}_{i \in \mathcal{I}_b}$. Further assume that $\boldsymbol{R}_{i,j}$'s are independent of each other, thus $e_i = \frac{1}{p}\sum_{j=1}^p \boldsymbol{R}_{i,j}$ is the sum of independent random variables with a common mean. Thus, according to the Kolmogorov Strong Law of Large Numbers (KSLLN), we know that $e_i$ converges to a constant almost surely, which in turn indicates that $v_i$ also converge some constant almost surely. The Theorem 1 summarizes the results formally, with the detailed proof provided in Appendix C.

**Theorem 1.** *Assuming $\{\boldsymbol{R}_{:,j}\}_{1 \le j \le p}$ are independent of each other, under the Gaussian attack, we have for each local node $i$ that*

$$\lim_{N^* \to \infty} \lim_{p \to \infty} e_i = \frac{n+1}{2} \ \ a.s., \tag{4}$$

$$\lim_{N^* \to \infty} \lim_{p \to \infty} \left( v_i - \bar{s}_b^2 \cdot \mathbb{I}(i \in \mathcal{I}_b) - \bar{s}_m^2 \cdot \mathbb{I}(i \in \mathcal{I}_m) \right) = 0 \ \ a.s., \tag{5}$$

*where $\mathbb{I}(\cdot)$ stands for the indicator function, $\bar{s}_b^2 \triangleq \frac{1}{p}\sum_{j=1}^p s_{b,j}^2$ and $\bar{s}_m^2 \triangleq \frac{1}{p}\sum_{j=1}^p s_{m,j}^2$.*

Considering that $\bar{s}_b^2 = \bar{s}_m^2$ if and only if $\Sigma_{j,j}$'s fall into a lower dimensional manifold whose measurement is zero under the Lebesgue measure, we have $P(\bar{s}_b^2 = \bar{s}_m^2) = 0$ if the attacker specifies the Gaussian variance $\Sigma_{j,j}$'s arbitrarily in the Gaussian attack. Thus, Theorem 1 in fact suggests that the benign nodes and the malicious nodes are different on the value of $v_i$, and therefore provides a guideline to detect the malicious nodes. Although the we do need $N^*$ and $p$ to go to infinity for getting the theoretical results in Theorem 1, in practice the malicious node detection algorithm based on the theorem typically works very well when $N^*$ and $p$ are reasonably large and $N_i$'s are not dramatically far away from each other.

The independent rank assumption in Theorem 1, which assumes that $\{\boldsymbol{R}_{:,j}\}_{1 \le j \le p}$ are independent of each other, may look restrictive. However, in fact it is a mild condition that can be easily satisfied in practice due to the following reasons. First, for a benign node $i \in \mathcal{I}_b$, $\boldsymbol{M}_{i,j}$ and $\boldsymbol{M}_{i,k}$ are often nearly independent, as the correlation between two model parameters $\boldsymbol{\theta}_j$ and $\boldsymbol{\theta}_k$ is often very weak in a larger deep neural network with a huge number of parameters. To verify the statement, we implemented independence tests for 100,000 column pairs randomly chosen from the message matrix $\boldsymbol{M}$ generated from the FASHION-MNIST data. Distribution of the p-values of these tests are demonstrated in Figure 2 via a histogram, which is very close to a uniform distribution, indicating that $\boldsymbol{M}_{i,j}$ and $\boldsymbol{M}_{i,k}$ are indeed nearly independent in practice. Second, even some $\boldsymbol{M}_{:,j}$ and $\boldsymbol{M}_{:,k}$ shows strong correlation, magnitude of the correlation would be reduced greatly during the transformation from $\boldsymbol{M}$ to $\boldsymbol{R}$, as the final ranking $\boldsymbol{R}_{i,j}$ also depends on many other factors. Actually, the independent rank assumption could be relaxed to be uncorrelated rank assumption which assumes the ranks are uncorrelated with each other. Adopting the weaker assumption will result in a change of convergence type of our theorems from the "almost surely convergence" to "convergence in probability", but with no essential influence to the our algorithm below.

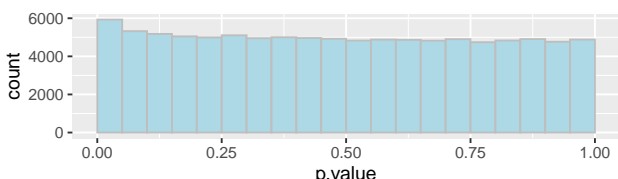

Figure 2: Independence tests for 100,000 column pairs randomly chosen from message matrix $\boldsymbol{M}$ generated from FASHION-MNIST data supports the independence assumption made in Theorem 1.

### 2.4 MALICIOUS NODE DETECTION FOR SIGN FLIPPING ATTACK

**Definition 2** (Sign flipping attack). *Sign flipping attack aims to generate the gradient values of malicious nodes by flipping the sign of the average of all the benign nodes' gradient at each epoch, i.e., specifying $\boldsymbol{M}_{i,:} = -r\boldsymbol{m}_{b,:}$ for any $i \in \mathcal{I}_m$, where $r > 0, \boldsymbol{m}_b = \frac{1}{n_1}\sum_{k \in \mathcal{I}_b} \boldsymbol{M}_{k,:}$.*

Based on the above definition, the update message of a malicious node $i$ under the sign flipping attack is

$$\boldsymbol{M}_{i,:} = -r\boldsymbol{m}_{b,:} = -\frac{r}{n_1} \sum_{k \in \mathcal{I}_b} \boldsymbol{M}_{k,:}. \tag{6}$$

For fixed $\{\boldsymbol{M}_{k,:}\}_{k \in \mathcal{I}_b}$, $\boldsymbol{M}_{i,:}$ is also a fixed vector without randomness, as it is a deterministic function of $\{\boldsymbol{M}_{k,:}\}_{k \in \mathcal{I}_b}$. On the other hand, however, we can also treat $\boldsymbol{M}_{i,:}$ as a random vector, since the randomness of $\{\boldsymbol{M}_{k,:}\}_{k \in \mathcal{I}_b}$ can be transferred to $\boldsymbol{M}_{i,:}$ via the link function in equation 6. In fact, for any parameter dimension $j$, considering that $\boldsymbol{M}_{k,j} \xrightarrow{d} \mathcal{N}\left(\mu_j, \sigma_j^2/N_k\right)$ for any $k \in \mathcal{I}_b$ according to Lemma 1, it is straightforward to see that $\boldsymbol{M}_{i,j} = -\frac{r}{n_1} \sum_{k \in \mathcal{I}_b} \boldsymbol{M}_{k,j}$ can also be well approximated by a Gaussian distribution. The lemma 3 summarizes the result formally.

**Lemma 3.** *Under the sign flipping attack, for each malicious node $i \in \mathcal{I}_m$ and any parameter dimension $j$, we have $\boldsymbol{M}_{i,j} = -\frac{r}{n_1} \sum_{k \in \mathcal{I}_b} \boldsymbol{M}_{k,j}$ is a deterministic function of $\{\boldsymbol{M}_{k,j}\}_{k \in \mathcal{I}_b}$, whose limiting distribution when $N^*$ goes to infinity is*

$$\boldsymbol{M}_{i,j} \xrightarrow{d} \mathcal{N}\left(\mu_j(r), \sigma_j^2(r)\right),\ 1 \le j \le p, \tag{7}$$

*where $\mu_j(r) = -r\mu_j$, $\sigma_j^2(r) = \frac{r^2 \cdot \sigma_j^2}{n_1 \cdot \bar{N}_b}$, and $\bar{N}_b = \frac{n_1}{\sum_{k \in \mathcal{I}_b} \frac{1}{N_k}}$ is the harmonic mean of $\{N_k\}_{k \in \mathcal{I}_b}$.*

Lemma 1 and Lemma 3 tell us that for each parameter dimension $j$, the distribution of $\{\boldsymbol{M}_{i,j}\}_{i=1}^n$ is a mixture of Gaussian components $\{\mathcal{N}\left(\mu_j, \sigma_j^2/N_i\right)\}_{i \in \mathcal{I}_b}$ centered at $\mu_j$ plus a point mass located at $\mu_j(r) = -r\mu_j$. If $N_i$'s are reasonably large, variances $\sigma_j^2/N_i$'s would be very close to zero, and the probability mass of the mixture distribution would concentrate to two local centers $\mu_j$ and $\mu_j(r) = -r\mu_j$, one for the benign nodes and the other one for the malicious nodes. This intuition provides us the guidance to identify the malicious nodes in this attack pattern. Transforming to the rank domain, the above intuition leads to different behavior patterns of the benign nodes and the malicious nodes in the rank matrix $\boldsymbol{R}$, which in turn result in different limiting behavior of $(e_i, v_i)$ for the benign and malicious nodes. The theorem 2 summarizes the results formally, with the detailed proof provided in Appendix D.

**Theorem 2.** *With the same independent rank assumption as posed in Theorem 1, under the sign flipping attack, we have for each local node $i$ that*

$$\lim_{N^* \to \infty} \lim_{p \to \infty} e_i = \bar{\mu}_b \cdot \mathbb{I}(i \in \mathcal{I}_b) + \bar{\mu}_m \cdot \mathbb{I}(i \in \mathcal{I}_m)\ a.s., \tag{8}$$

$$\lim_{N^* \to \infty} \lim_{p \to \infty} v_i = \bar{s}_b^2 \cdot \mathbb{I}(i \in \mathcal{I}_b) + \bar{s}_m^2 \cdot \mathbb{I}(i \in \mathcal{I}_m)\ a.s., \tag{9}$$

*where $\bar{\mu}_b = \frac{n+n_0+1}{2} - n_0\rho$, $\bar{\mu}_m = n_1\rho + \frac{n_0+1}{2}$, $\rho = \lim_{p \to \infty} \frac{\sum_{j=1}^p \mathbb{I}(\mu_j > 0)}{p}$, $\bar{s}_m^2$ and $\bar{s}_b^2$ are both quadratic functions of $\rho$ whose concrete form also depends on $n_0$ and $n_1$.*

Considering that $\bar{\mu}_b = \bar{\mu}_m$ if and only if $\rho = \frac{1}{2}$, and $\bar{s}_b^2 = \bar{s}_m^2$ if and only if $\rho$ is the solution of a quadratic function, the probability of $(\bar{\mu}_b, \bar{s}_b^2) = (\bar{\mu}_m, \bar{s}_m^2)$ is zero as $p \to \infty$. Such a phenomenon suggests that we can detect the malicious nodes based on the moments $(e_i, v_i)$ to defense the sign flipping attack as well. Noticeably, we note that the limit behaviour of $e_i$ and $v_i$ does not dependent on the specification of $r$, which defines the sign flipping attack. Although such a fact looks a bit abnormal at the first glance, it is totally understandable once we realize that with the variance of $\boldsymbol{M}_{i,j}$ shrinks to zero with $N_i$ goes to infinity for each benign node $i$, any different between $\mu_j$ and $\mu_j(r)$ would result in the same rank vector $\boldsymbol{R}_{:,j}$ in the rank domain.

## 2.5 Malicious node detection for zero gradient attack

**Definition 3** (Zero gradient attack). *Zero gradient attack aims to make the aggregated message to be zero, i.e., $\sum_{i=1}^n \boldsymbol{M}_{i,:} = 0$, at each epoch, by specifying $\boldsymbol{M}_{i,:} = -\frac{n_1}{n_0}\boldsymbol{m}_{b,:}$ for all $i \in \mathcal{I}_m$.*

Apparently, the zero gradient attack defined above is a special case of sign flipping attack by specifying $r = \frac{n_1}{n_0}$. Since the conclusions of Theorem 2 keep unchanged for different specifications of $r$ as we have discussed, we have the following corollary for zero gradient attack.

**Corollary 1.** *Under the zero gradient attack, $e_i$'s and $v_i$'s follow exactly the same limiting behaviours as described in Theorem 2.*

## 2.6 MANDERA

Theorem 1, 2 and Corollary 1 imply that, under these three attacks (Gaussian attack, zero gradient attack and sign flipping attack), the first two moments of $\boldsymbol{R}_{i,:}$, i.e., $(e_i, v_i)$, converge to two different limits for the benign nodes and the malicious nodes, respectively. Thus, for a real dataset where $N_i$'s and $p$ are all finite but reasonably large numbers, the scatter plot of $\{(e_i, v_i)\}_{1 \leq i \leq n}$ would demonstrate a clustering structure: one cluster for the benign nodes and the other cluster for the malicious nodes. Figure 3 illustrates such a scatter plot for the 100 local nodes in a typical epoch of training the FASHION-MNIST dataset under different FL settings (to keep the two dimensions of the scatter plot to the same scale, we replaced $v_i$ by its square root $s_i = \sqrt{v_i}$ instead). Clearly, a simple clustering procedure would detect the malicious nodes from the scatter plot. Based on this intuition, we propose *MAlicious Node DEtection via RAnking* (MANDERA) to detect the malicious nodes, whose workflow is detailed in Algorithm 1.

---

**Algorithm 1** Malicious node detection via ranking (MANDERA)

---

**Input:** The message matrix $\boldsymbol{M}$.
1: Convert the message matrix $\boldsymbol{M}$ to the ranking matrix $\boldsymbol{R}$ by applying *Rank* operator.
2: Compute mean and standard deviation of rows in $\boldsymbol{R}$, i.e., $\{(e_i, s_i)\}_{1 \leq i \leq n}$.
3: Run the clustering algorithm $K$-means to $\{(e_i, s_i)\}_{1 \leq i \leq n}$ with $K = 2$, and predict the set of benign nodes with the lager cluster denoted by $\hat{\mathcal{I}}_b$.

**Output:** The predicted benign node set $\hat{\mathcal{I}}_b$.

---

**Remark.** *MANDERA can be applied to either a single epoch or multiple epochs. For a single-epoch mode, the input data $\boldsymbol{M}$ is the message matrix received from a single epoch. For multiple-epoch mode, the data $\boldsymbol{M}$ is the column-concatenation of the message matrices from multiple epochs. By default, the experiments below all use single epoch to detect the malicious nodes.*

The predicted benign nodes $\hat{\mathcal{I}}_b$ obtained by MANDERA naturally leads to an aggregated message $\hat{\boldsymbol{m}}_{b,:} = \frac{1}{\#(\hat{\mathcal{I}}_b)} \sum_{i \in \hat{\mathcal{I}}_b} \boldsymbol{M}_{i,:}$. The theorem 3 shows that $\hat{\mathcal{I}}_b$ and $\hat{\boldsymbol{m}}_b$ lead to consistent estimations of $\mathcal{I}_b$ and $\boldsymbol{m}_b$ respectively, indicating that MANDERA enjoys *robustness guarantee* (Steinhardt, 2018) for typical Byzantine attacks.

**Theorem 3.** *Under the three typical Byzantine attacks, i.e., Gaussian attack, sign flipping attack and zero gradient attack, we have:*

$$\lim_{N^* \to \infty} \lim_{p \to \infty} \mathbb{P}(\hat{\mathcal{I}}_b = \mathcal{I}_b) = 1 \quad and \quad \lim_{N^* \to \infty} \lim_{p \to \infty} \mathbb{E}||\hat{\boldsymbol{m}}_{b,:} - \boldsymbol{m}_{b,:}||_2 = 0. \tag{10}$$

The proof of Theorem 3 can be found in Appendix E.

## 3 EXPERIMENTS

We evaluate the efficacy in detecting malicious nodes within the federated learning framework with the use of three Datasets. The first is the FASHION-MNIST dataset (Xiao et al., 2017), a dataset of 60,000 and 10,000 training and testing samples respectively divided into 10 classes of apparel. The second is CIFAR-10 (Krizhevsky et al., 2009), a dataset of 60,000 small object images also containing 10 object classes. The third is MNIST (Deng, 2012) dataset which appears in Appendix H. In these experiments we mainly adopt implementations of Byzantine attacks released by (Wu et al., 2020b;a) and the label flipping attack (Tolpegin et al., 2020b;a). The label flipping attack is a data poisoning attack that alters one or more labels of training data to an attacker's pre-determined target label. For example, in CIFAR-10's object labels, an attacker may change the labels of their local *cat* images to be labelled as *dogs*. We use the Label Flipping attack as a comparative poisoning attack that achieves its objective in a more subtle manner. In our experiments, we set $\Sigma = 30\boldsymbol{I}$ for the Gaussian attack and $r = 3$ for the sign flipping attack, where $\boldsymbol{I}$ is the identity matrix. For all experiments we fix $n = 100$ participating nodes, of which a variable number of nodes are poisoned $|n_0| \in \{5, 10, 15, 20, 25, 30\}$. The training process is run until 25 epochs have elapsed. We have described the structure of these networks in Appendix A.

### 3.1 ILLUSTRATION OF THE AVERAGE RANKING AND STANDARD DEVIATION OF RANKING

Section 2 speculated that the distribution of parameter ranks differ sufficiently for the detection of malicious and benign nodes. We validate this hypothesis in Figure 3 by illustrating the difference

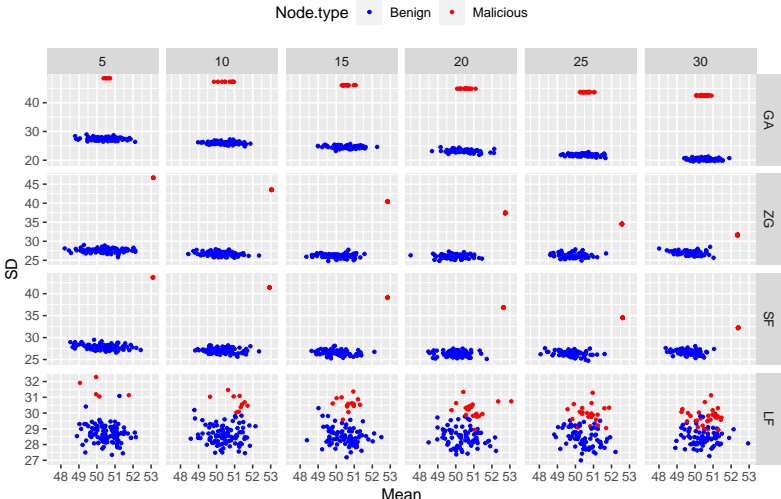

Figure 3: The scatter plots of $(e_i, s_i)$ for the 100 nodes under four types of attack as illustrative examples demonstrating ranking mean and variance from the 1st epoch of training for the FASHION-MNIST dataset.

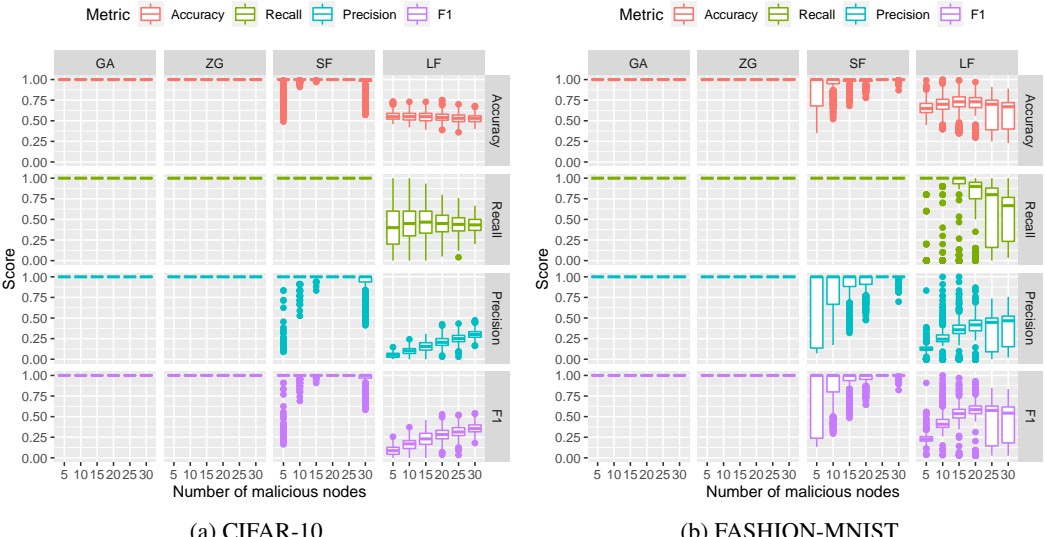

(a) CIFAR-10      (b) FASHION-MNIST

Figure 4: Classification performance of our proposed approach MANDERA (Algorithm 1) under four types of attack for CIFAR-10 and FASHION-MNIST data. Gaussian Attack (GA); Zero-Gradient (ZG); Sign-Flipping (SF); and Label-Flipping (LF). The boxplot bounds the 25th (Q1) and 75th (Q3) percentile, with the central line representing the 50th quantile (median). The end points of the whisker represent the Q1-1.5(Q3-Q1) and Q3+1.5(Q3-Q1) respectively.

between the benign nodes and malicious nodes in terms of the mean of gradients' rankings and the standard deviation of gradients' ranking.

It can be observed from Figure 3 that, under Gaussian and Label flipping attacks, the average rankings of malicious nodes are of a similar distribution to benign nodes. It is problematic for distinguishing between the two types of nodes, if only average ranking information is used. On the other hand, Figure 3 displays a larger separation of distributions for the standard deviation of ranking. It is noted that all 4 attacks observe a convergence of the distributions as the number of malicious nodes increase, increasing the difficulty of defense for both MANDERA and all other defenses. However, the likelihood of an attacker controlling increasingly large numbers of malicious nodes also decrease.

### 3.2 MALICIOUS NODE DETECTION BY MANDERA

We test the performance of MANDERA on the update gradients of a model under attacks. In this section, MANDERA acts as an observer without intervening in the learning process to identify malicious nodes with a set of gradients from a single epoch. Each configuration of 25 training epochs, with a given number of malicious nodes was repeated 20 times. Figure 4 demonstrates the classification performance (Metrics defined in Appendix B) of MANDERA with different settings of participating malicious nodes and the four poisoning attacks of Guassian Attack (GA), Zero Gradient attack (ZG), Sign Flipping attack (SF) and the Label Flipping attack (LF).

While we have formally demonstrated the efficacy of MANDERA in accurately detecting potentially malicious nodes participating in the federated learning process. In practice, to leverage an unsupervised K-means clustering algorithm, we must also identify the correct group of nodes as the malicious group. Our strategy is to identify the group with the most exact gradients, or otherwise the smaller group (we regard a system with over 50% of their nodes compromised as having larger issues than just poisoning attacks) [1]. We also test other clustering algorithms, such as hierarchical clustering and Gaussian mixture models (Fraley & Raftery, 2002). It turns out that the performance of MANDERA is quite robust with different choices of clustering methods. Detailed results can be found in Appendix F.

From Figure 4, it is immediately evident that the recall of the malicious nodes for the Byzantine attacks is exceptional. However, occasionally benign nodes have also been misclassified as malicious under a SF, and to a lesser extent the ZG attack for both datasets. On all attacks, in the presence of more malicious nodes, the recall of malicious nodes trends down. As for the data poisoning attack of LF, it is consistently more difficult to detect, however we note that the LF attack has a more subtle influence on the model in contrast to the impact of Byzantine attacks.

### 3.3 MANDERA FOR DEFENDING AGAINST POISONING ATTACKS

In this section, we encapsulate MANDERA into a module prior to the the aggregation step, MANDERA has the sole objective of identifying malicious nodes, and excluding their updates from the global aggregation step. Each configuration of 25 training epochs, a given poisoning attack, defense method, and a given number of malicious nodes was repeated 10 times. We compare MANDERA against 5 other robust aggregation defense methods, Krum (Blanchard et al., 2017), Bulyan (Guerraoui et al., 2018), Trimmed Mean (Yin et al., 2018), Median (Yin et al., 2018) and FLTrust (Cao et al., 2020). Of which the first 2 requires an assumed number of malicious nodes, and the latter 3 only aggregate robustly.

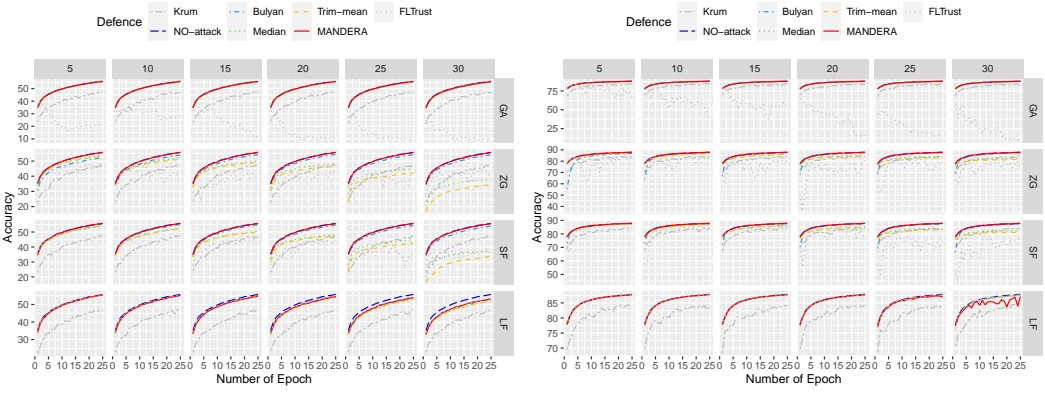

(a) CIFAR-10 Dataset  (b) FASHION-MNIST dataset

Figure 5: Model Accuracy at each epoch of training, each line of the curve represents a different defense against the poisoning attacks.

From Figure 5, it is observed that MANDERA performs about the same as the best performing defense mechanisms, close to the performance of a model not under attack. MANDERA's accuracy

---

[1]More informed approaches to selecting the malicious cluster can be tested in future work. E.g. Figure 3 displays less variation of rank variance in malicious cluster compared to benign nodes. This could robust selection of the malicious group, and enabling selection of malicious groups larger than 50%.

is observed to vary slightly under the LF attack on fashion data with 30 malicious nodes, this is consistent with the larger accuracy ranges previously observed in Figure 4b. Interestingly, FLTrust as a standalone defense is weak in protecting against the most extreme Byzantine attacks. However, we highlight that FLtrust is a robust aggregation method against targeted attacks that may thwart defences like Krum, Trimmed mean. We see FLTrust as a complementary defence that relies on a base method of defence against Byzantine attacks, but expands the protection coverage of the FL system against adaptive attacks.

### 3.4 COMPUTATIONAL EFFICIENCY

We have previously been able to observe that MANDERA can perform at par with the current highest performing poisoning attack defenses. Another benefit arises with the simplification of the mitigation strategy with the introduction of ranking at the core of the algorithm. Sorting and Ranking algorithms are fast. Additionally, we only apply clustering on the two dimensions of rank mean and standard deviation, in contrast to other works that seek to cluster on the entire node update (Chen et al., 2021). The times in Table 1 for MANDERA, Krum and Bulyan do not include the parameter/gradient aggregation step. These times were computed on 1 core of a Dual Xeon 14-core E5-2690, with 8 Gb of system RAM and a single Nvidia Tesla P100. Table 1 demonstrates that MANDERA is able to achieve a faster speed than that of single Krum [2] (by more than half) and Bulyan (by an order of magnitude).

Table 1: Mean and standard deviation of computational times for defense function given the same set of gradients from 100 nodes, of which 30 were malicious. Each function was repeated 100 times.

| Defense (Detection) | Mean ± SD (ms) | Defense (Aggregation) | Mean ± SD (ms) |
|---|---|---|---|
| *MANDERA* | 643 ± 8.646 | Trimmed Mean | 3.96 ± 0.41 |
| Krum (Single) | 1352 ± 10.09 | Median | 9.81 ± 3.88 |
| Bulyan | 27209 ± 233.4 | FLTrust | 361 ± 4.07 |

## 4 DISCUSSION AND CONCLUSION

If attackers create more adaptive attacks unlike Definition 1, 2 and 3, they may evade MANDERA and achieve model poisoning. In this work, we have configured our Federated Learner to use all 100 nodes in the learning process at every round, we acknowledge FL framework may learn the global model only using subset of nodes at each round. In these settings MANDERA would still function, as we would rank and cluster on the parameters of the participating nodes, without assuming any number of poisoned nodes. In Algorithm 1, performance could be improved by incorporating higher order moments. MANDERA is unable to function when gradients are securely aggregated in its current form. However, malicious nodes can be identified and excluded from the secure aggregation step, while still protecting the privacy of participating nodes by performing MANDERA through secure ranking (Zhang et al., 2013; Lin & Tzeng, 2005) (recall that MANDERA only requires the ranking matrix to detect poisoned nodes). It remains to be seen the effectiveness of MANDERA on more advanced poisoning techniques like adversarial poisoning or Evasion attacks.

In conclusion, we have provided theoretical guarantees and experimentally shown efficacy in the use of ranking algorithms for the detection of malicious nodes performing poisoning attacks against federated learning. Our proposed method MANDERA, is able to achieve high detection accuracy and maintain a model accuracy on par with other seminal, high performing defense mechanisms, but with three notable advantages. First, provable guarantees for the use of ranking to detect Gaussian, Zero Gradient and Sign Flipping attacks. Next, faster detection with the use of ranking algorithms. Finally, the MANDERA defense does not need a prior estimation of the number of poisoned nodes. In this work we demonstrate how the rank domain can be useful in applications to defend against malicious actors.

## Ethics Statement

The core objective of our research is to provide an additional means of defense against poisoning nodes that target Federated Learning. To test our defense we have implemented different attacks against the Federated Learning framework. Attackers may adopt our defense strategy to design new poisoning attacks. Fortunately, these poisoning attacks can not be leveraged to leak private information from Federated learning models, instead only impact its performance.

---

[2]The use of multi-krum would have yielded better protection (c.f. Section 3) at the behest of speed.

## Reproducibility Statement

To ensure reproducible research, we have supplemented our proposal for MANDERA, by supplying both R and Python implementations of MANDERA used in this paper, uploaded with the remainder of the experiment code. The three datasets featured in this paper is CIFAR-10 (Krizhevsky et al., 2009), Fasion-MNIST (Xiao et al., 2017), and MNIST (Deng, 2012); we have used each of these dataset unaltered from their respective sources. We have stated the assumptions in our theorems and their proofs can be found in the Appendix. But to explain our assumptions in simple terms, (1) The data samples on each local node are independently drawn from the same distribution. (2) The gradient value for each parameter is independent to each other.

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

## A  NEURAL NETWORK CONFIGURATIONS

We train these models with a batch size of 10, an SGD optimizer operates with a learning rate of 0.01, and 0.5 momentum for 25 epochs. The accuracy of the model is evaluated on a holdout set of 1000 samples.

### A.1  FASHION-MNIST AND MNIST

- Layer 1: $1 * 16 * 5$, 2D Convolution, Batch Normalization, ReLU Activation, Max pooling.
- Layer 2: $16 * 32 * 5$, 2D Convolution, Batch Normalization, ReLU Activation, Max pooling.
- Output: 10 Classes, Linear.

### A.2  CIFAR-10

- Layer 1: $1 * 32 * 3$, 2D Convolution, Batch Normalization, ReLU Activation, Max pooling.
- Layer 2: $32 * 32 * 3$, 2D Convolution, Batch Normalization, ReLU Activation, Max pooling.
- Output: 10 Classes, Linear.

## B    METRICS

The metrics observed in Section 3 to evaluate the performance of the defense mechanisms are defined as follows:

$$\text{Precision} = \frac{\text{TP}}{\text{TP+FP}},$$
$$\text{Accuracy} = \frac{\text{TP+TN}}{\text{TP+FP+FN+TN}},$$
$$\text{Recall} = \frac{\text{TP}}{\text{TP+FN}},$$
$$\text{F1} = 2 \times \frac{\text{Precision} \times \text{Recall}}{\text{Precision+Recall}}.$$

## C    PROOF OF THEOREM 1

*Proof.* Because $\{\boldsymbol{R}_{i,j}\}_{1 \leq j \leq p}$ are independent random variables with a finite upper bound (since $n$ is fixes) as assumed, direct application of KSLLN leads to

$$\lim_{p \to \infty} \frac{1}{p} \sum_{j=1}^{p} \left( \boldsymbol{R}_{i,j} - \mathbb{E}(\boldsymbol{R}_{i,j}) \right) = 0 \ a.s., \tag{11}$$

$$\lim_{p \to \infty} \frac{1}{p} \sum_{j=1}^{p} \left[ \left( \boldsymbol{R}_{i,j} - \mathbb{E}(\boldsymbol{R}_{i,j}) \right)^2 - \text{Var}(\boldsymbol{R}_{i,j}) \right] = 0 \ a.s.. \tag{12}$$

To prove Theorem 1 based on Equation 11 and 12, we need to derive the concrete form of $\mathbb{E}(\boldsymbol{R}_{i,j})$ and $\text{Var}(\boldsymbol{R}_{i,j})$.

Fortunately, because $\boldsymbol{M}_{i,j} \xrightarrow{d} \mathcal{N}(\mu_j, \Sigma_{j,j})$ for $\forall \, i \in \mathcal{I}_m$ and $\boldsymbol{M}_{i,j} \xrightarrow{d} \mathcal{N}(\mu_j, \sigma_j^2/N_i)$ for $\forall \, i \in \mathcal{I}_b$ when $N^* \to \infty$, it is straightforward to see due to the symmetry of Gaussian distribution that

$$\lim_{N^* \to \infty} \mathbb{E}(\boldsymbol{R}_{i,j}) = \frac{n+1}{2}, \ 1 \leq i \leq n, \ 1 \leq j \leq p. \tag{13}$$

Moreover, assuming that the sample sizes of different benign nodes approach to each other with $N^*$ going to infinity, i.e.,

$$\lim_{N^* \to \infty} \frac{1}{N^*} \max_{i,k \in \mathcal{I}_b} |N_i - N_k| = 0, \tag{14}$$

for each parameter dimension $j$, $\{\boldsymbol{M}_{i,j}\}_{i \in \mathcal{I}_b}$ would converge to the same Gaussian distribution $\mathcal{N}(\mu_j, \sigma_j^2/N^*)$ with the increase of $N^*$. Thus, due to the exchangeability of $\{\boldsymbol{M}_{i,j}\}_{i \in \mathcal{I}_b}$ and $\{\boldsymbol{M}_{i,j}\}_{i \in \mathcal{I}_m}$, it is easy to see that there exist two positive constants $s_b^2$ and $s_m^2$, such that

$$\lim_{N^* \to \infty} \text{Var}(\boldsymbol{R}_{i,j}) = s_b^2 \cdot \mathbb{I}(i \in \mathcal{I}_b) + s_{m,j}^2 \cdot \mathbb{I}(i \in \mathcal{I}_m), \tag{15}$$

where $s_{b,j}^2$ and $s_{m,j}^2$ are both complex functions of $n_0$, $n_1$, $\sigma_j^2$, $\Sigma_{j,j}$ and $N^*$, and $s_{b,j}^2 = s_{m,j}^2$ if and only if $\sigma_j^2/N^* = \Sigma_{j,j}$.

Combining Equation 11 and 13, we have

$$\lim_{N^* \to \infty} \lim_{p \to \infty} e_i = \lim_{N^* \to \infty} \lim_{p \to \infty} \frac{1}{p} \sum_{j=1}^{p} \boldsymbol{R}_{i,j} = \frac{n+1}{2} \ a.s.,$$

i.e., Equation 4, which further indicates that $e_i$ and $\mathbb{E}(\boldsymbol{R}_{i,j})$ share the same limit when both $p$ and $N^*$ go to infinity. Thus, we have

$$\lim_{N^* \to \infty} \lim_{p \to \infty} v_i = \lim_{N^* \to \infty} \lim_{p \to \infty} \frac{1}{p} \sum_{j=1}^{p} \left( \boldsymbol{R}_{i,j} - e_i \right)^2$$

$$= \lim_{N^* \to \infty} \lim_{p \to \infty} \frac{1}{p} \sum_{j=1}^{p} \left( \boldsymbol{R}_{i,j} - \mathbb{E}(\boldsymbol{R}_{i,j}) \right)^2 \ a.s.. \tag{16}$$

Combining Equation 12, 15, and 16, we have

$$\lim_{N^* \to \infty} \lim_{p \to \infty} \left( v_i - \bar{s}_b^2 \cdot \mathbb{I}(i \in \mathcal{I}_b) - \bar{s}_m^2 \cdot \mathbb{I}(i \in \mathcal{I}_m) \right) = 0 \ a.s.,$$

i.e., Equation 5. Thus, the proof is complete. □

## D  PROOF OF THEOREM 2

*Proof.* It is straightforward to see that equation 11 also holds for sign flipping attack under the assumptions of Theorem 2. But, we need to re-calculate $\mathbb{E}(\boldsymbol{R}_{i,j})$ for benign and malicious nodes under the new setting.

Under the sign flipping attack, because $\boldsymbol{M}_{i,j} \xrightarrow{d} \mathcal{N}\left(\mu_j(r), \sigma_j^2(r)\right)$ for $\forall\, i \in \mathcal{I}_m$ and $\boldsymbol{M}_{i,j} \xrightarrow{d} \mathcal{N}\left(\mu_j, \sigma_j^2/N_i\right)$ for $\forall\, i \in \mathcal{I}_b$ when $N^* \to \infty$, and

$$\lim_{N^* \to \infty} (\sigma_j^2/N_i) = \lim_{N^* \to \infty} \sigma_j^2(r) = 0,$$

it is straightforward to see that

$$\lim_{N^* \to \infty} P(\boldsymbol{M}_{i,j} > \boldsymbol{M}_{k,j}) = \mathbb{I}(\mu_j > 0), \ \forall\, i \in \mathcal{I}_b, \forall\, k \in \mathcal{I}_m,$$

which further indicates that

$$
\begin{aligned}
\lim_{N^* \to \infty} \mathbb{E}(\boldsymbol{R}_{i,j}) &= \frac{n_1 + 1}{2} \cdot \mathbb{I}(i \in \mathcal{I}_b) + \frac{n + n_1 + 1}{2} \cdot \mathbb{I}(i \in \mathcal{I}_m) \ if \ \mu_j > 0, \\
\lim_{N^* \to \infty} \mathbb{E}(\boldsymbol{R}_{i,j}) &= \frac{n_0 + 1}{2} \cdot \mathbb{I}(i \in \mathcal{I}_m) + \frac{n + n_0 + 1}{2} \cdot \mathbb{I}(i \in \mathcal{I}_b) \ if \ \mu_j < 0;
\end{aligned}
\tag{17}
$$

$$
\begin{aligned}
\lim_{N^* \to \infty} \mathbb{E}(\boldsymbol{R}_{i,j}^2) &= S_{[1,n_1]}^2 \cdot \mathbb{I}(i \in \mathcal{I}_b) + S_{[n_1+1,n]}^2 \cdot \mathbb{I}(i \in \mathcal{I}_m) \ if \ \mu_j > 0, \\
\lim_{N^* \to \infty} \mathbb{E}(\boldsymbol{R}_{i,j}^2) &= S_{[1,n_0]}^2 \cdot \mathbb{I}(i \in \mathcal{I}_m) + S_{[n_0+1,n]}^2 \cdot \mathbb{I}(i \in \mathcal{I}_b) \ if \ \mu_j < 0,
\end{aligned}
\tag{18}
$$

where $S_{[a,b]}^2 = \frac{1}{b-a+1} \sum_{k=a}^{b} k^2$.

Combining Equation 11 and 17, we have

$$
\lim_{N^* \to \infty} \lim_{p \to \infty} e_i =
\begin{cases}
\rho \cdot \frac{n+n_1+1}{2} + (1 - \rho) \cdot \frac{n_0+1}{2} = \bar{\mu}_m \ a.s., & \text{if } i \in \mathcal{I}_m, \\
\rho \cdot \frac{n_1+1}{2} + (1 - \rho) \cdot \frac{n+n_0+1}{2} = \bar{\mu}_b \ a.s., & \text{if } i \in \mathcal{I}_b,
\end{cases}
$$

where $\rho = \lim_{p \to \infty} \frac{\sum_{j=1}^{p} \mathbb{I}(\mu_j > 0)}{p}$, i.e., Equation 8.

Define $\bar{\mu}_i = \bar{\mu}_m \cdot \mathbb{I}(i \in \mathcal{I}_m) + \bar{\mu}_b \cdot \mathbb{I}(i \in \mathcal{I}_b)$. Based on KSLLN, we have:

$$\lim_{p \to \infty} \frac{1}{p} \sum_{j=1}^{p} \left[ (\boldsymbol{R}_{i,j} - \bar{\mu}_i)^2 - \mathbb{E}(\boldsymbol{R}_{i,j} - \bar{\mu}_i)^2 \right] = 0 \ a.s..$$

As we have proved in Equation 8 that

$$\lim_{N^* \to \infty} \lim_{p \to \infty} e_i = \bar{\mu}_i \ a.s.,$$

we have

$$\lim_{N^* \to \infty} \lim_{p \to \infty} \frac{1}{p} \sum_{j=1}^{p} \left[ (\boldsymbol{R}_{i,j} - e_i)^2 - \mathbb{E}(\boldsymbol{R}_{i,j} - \bar{\mu}_i)^2 \right] = 0 \ a.s.,$$

which implies that

$$\lim_{N^* \to \infty} \lim_{p \to \infty} v_i = \lim_{N^* \to \infty} \lim_{p \to \infty} \frac{1}{p} \sum_{j=1}^{p} (\boldsymbol{R}_{i,j} - e_i)^2 = \lim_{N^* \to \infty} \lim_{p \to \infty} \frac{1}{p} \sum_{j=1}^{p} \mathbb{E}(\boldsymbol{R}_{i,j} - \bar{\mu}_i)^2 \ a.s..$$

Considering that

$$\lim_{N^* \to \infty} \lim_{p \to \infty} \frac{1}{p} \sum_{j=1}^{p} \mathbb{E}(\boldsymbol{R}_{i,j} - \bar{\mu}_i)^2$$

$$= \lim_{p \to \infty} \lim_{N^* \to \infty} \frac{1}{p} \sum_{j=1}^{p} \left( \mathbb{E}(\boldsymbol{R}_{i,j}^2) - 2\bar{\mu}_i \mathbb{E}(\boldsymbol{R}_{i,j}) + (\bar{\mu}_i)^2 \right)$$

$$= \left[ \bar{\tau}_m - (\bar{\mu}_m)^2 \right] \cdot \mathbb{I}(i \in \mathcal{I}_m) + \left[ \bar{\tau}_b - (\bar{\mu}_b)^2 \right] \cdot \mathbb{I}(i \in \mathcal{I}_b),$$

where

$$\bar{\tau}_b = \rho \cdot S_{[1,n_1]}^2 + (1 - \rho) \cdot S_{[n_0+1,n]}^2,$$
$$\bar{\tau}_m = \rho \cdot S_{[n_1+1,n]}^2 + (1 - \rho) \cdot S_{[1,n_0]}^2.$$

we have

$$\lim_{N^* \to \infty} \lim_{p \to \infty} v_i = \left[ \bar{\tau}_m - (\bar{\mu}_m)^2 \right] \cdot \mathbb{I}(i \in \mathcal{I}_m) + \left[ \bar{\tau}_b - (\bar{\mu}_b)^2 \right] \cdot \mathbb{I}(i \in \mathcal{I}_b).$$

It completes the proof of Equation 9 by specifying $\bar{s}_b^2 = \bar{\tau}_b - (\bar{\mu}_b)^2$ and $\bar{s}_m^2 = \bar{\tau}_m - (\bar{\mu}_m)^2$. $\qquad\square$

## E    PROOF OF THEOREM 3

*Proof.* According to Theorem 1, 2 and Corollary 1, when both $N^*$ and $p$ are large enough, with probability 1 there exist $(e_b, v_b)$, $(e_m, v_m)$ and $\delta > 0$ such that $||(e_b, v_b) - (e_m, v_m)||_2 > \delta$, and

$$||(e_i, v_i) - (e_b, v_b)||_2 \le \frac{\delta}{2} \text{ for } \forall\, i \in \mathcal{I}_b \quad \text{and} \quad ||(e_i, v_i) - (e_m, v_m)||_2 \le \frac{\delta}{2} \text{ for } \forall\, i \in \mathcal{I}_m.$$

Therefore, with a reasonable clustering algorithm such as $K$-mean with $K = 2$, we would expect $\hat{\mathcal{I}}_b = \mathcal{I}_b$ with probability 1.

Because we can always find a $\Delta > 0$ such that $||\boldsymbol{M}_{i,:} - \boldsymbol{M}_{j,:}||_2 \le \Delta$ for any node pair $(i, j)$ in a fixed dataset with a finite number of nodes, and $\hat{\boldsymbol{m}}_{b,:} = \boldsymbol{m}_{b,:}$ when $\hat{\mathcal{I}}_b = \mathcal{I}_b$, we have

$$\mathbb{E}||\hat{\boldsymbol{m}}_{b,:} - \boldsymbol{m}_{b,:}||_2 \le \Delta \cdot \mathbb{P}(\hat{\mathcal{I}}_b \ne \mathcal{I}_b),$$

and thus

$$\lim_{N^* \to \infty} \lim_{p \to \infty} \mathbb{E}||\hat{\boldsymbol{m}}_{b,:} - \boldsymbol{m}_{b,:}||_2 = 0.$$

It completes the proof. $\qquad\square$

## F    MANDERA PERFORMANCE WITH DIFFERENT CLUSTERING ALGORITHMS

In this section, Figure 6 demonstrate that the discriminating performance of MANDERA when hierarchical clustering and Gaussian mixture models are used in-place of K-means for FASHION-MNIST data set remain robust.

## G    MODEL LOSSES ON CIFAR-10 AND FASHION-MNIST DATA

Figure 7 presents the model loss to accompany the model prediction performance of Figure 5 previously seen in Section 3.

## H    MODEL PERFORMANCE ON MNIST DATA

In this section we replicate experiments that were previously performed in Section 3 on the MNIST (Deng, 2012) dataset. The MNIST dataset is a dataset of 60,000 and 10,000 training and testing samples respectively divided into 10 classes of handwritten digits from multiple authors. Figure 8 contains the performance characteristics of MANDERA's defense against the four attacks, whilst Figure 9 contains the comparative accuracy and loss when the different defenses are applied. Generally speaking, the observations previously observed continue to hold for this dataset.

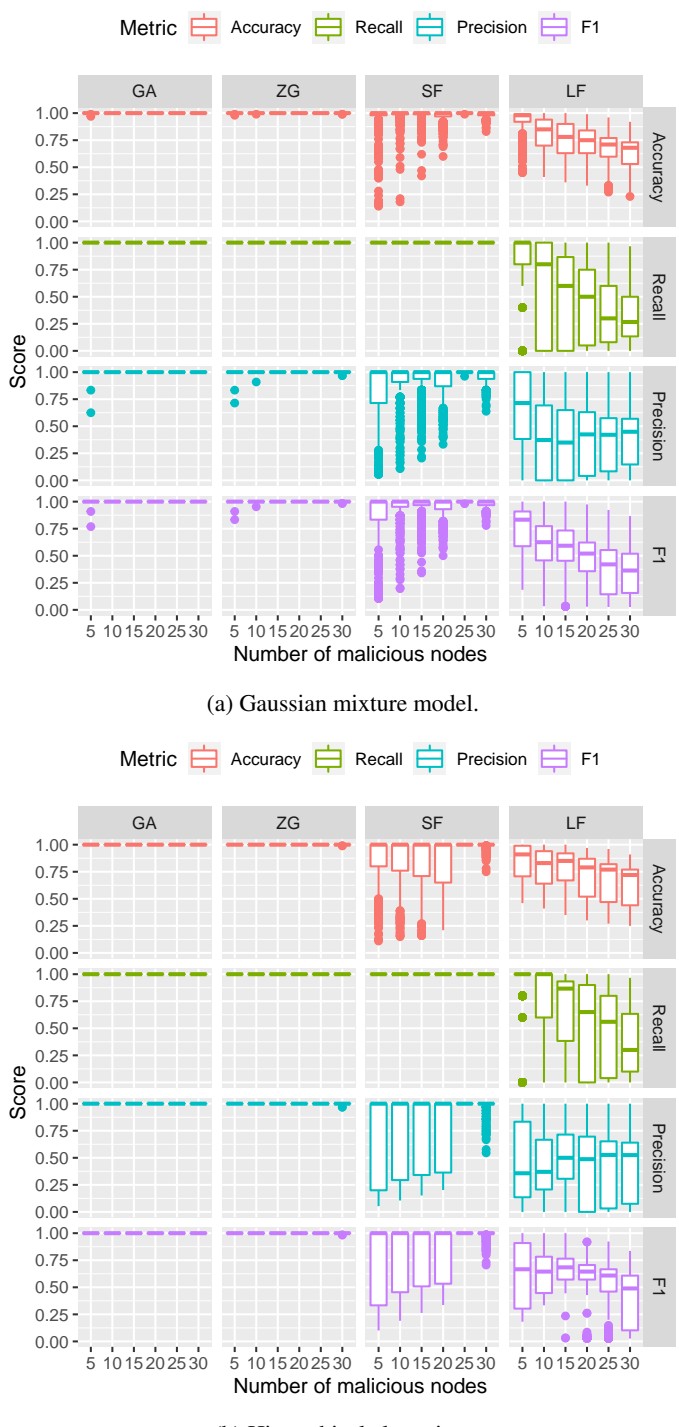

(a) Gaussian mixture model.

(b) Hierarchical clustering.

Figure 6: Classification performance of our proposed approach MANDERA (Algorithm 1) with other clustering algorithms under four types of attack for FASHION-MNIST data. GA: Gaussian attack; ZG: Zero-gradient attack; SF: Sign-flipping; and LF: Label-flipping. The boxplot bounds the 25th (Q1) and 75th (Q3) percentile, with the central line representing the 50th quantile (median). The end points of the whisker represent the Q1-1.5(Q3-Q1) and Q3+1.5(Q3-Q1) respectively.

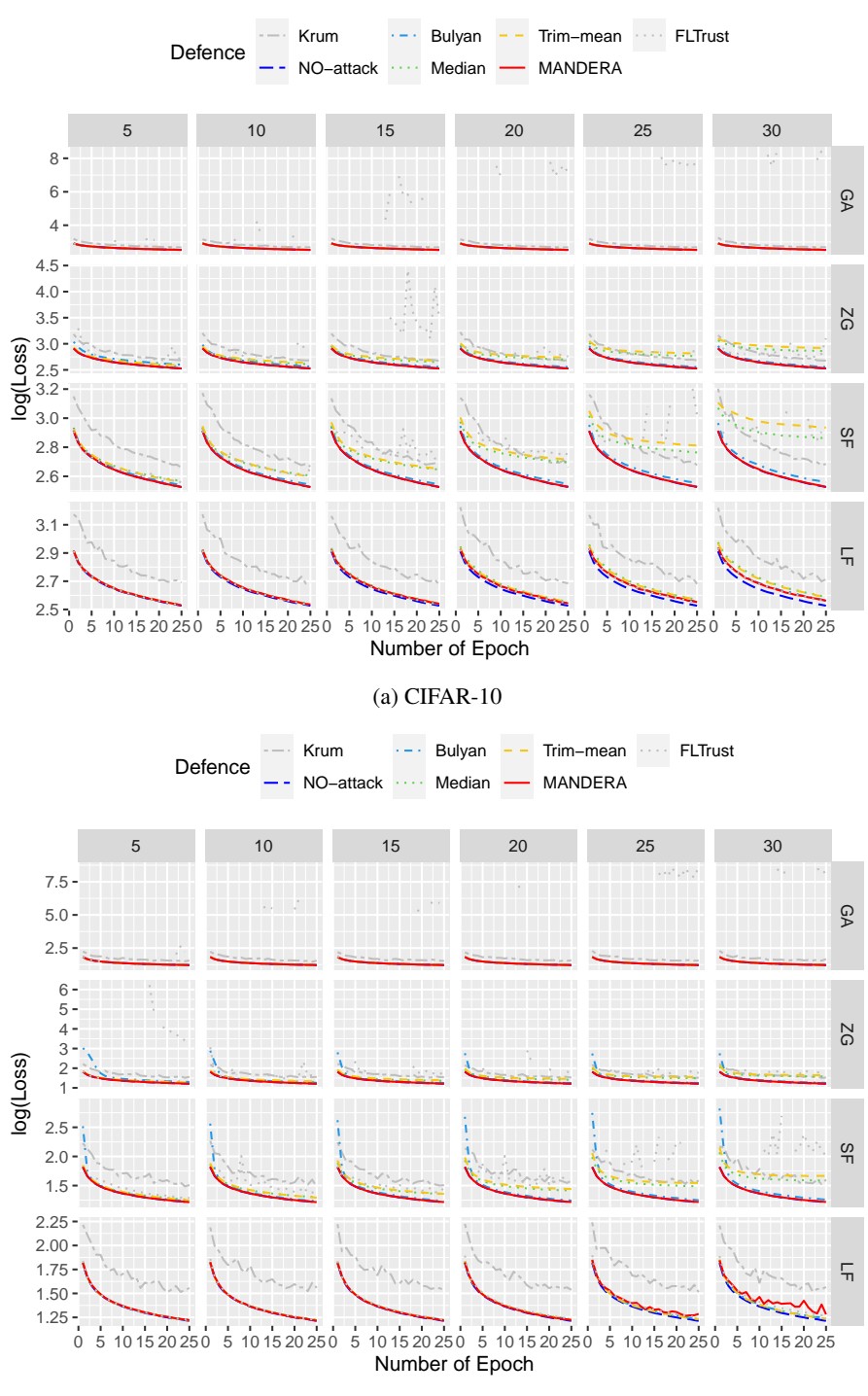

(a) CIFAR-10

(b) FASHION-MNIST

Figure 7: Model Loss at each epoch of training, each line of the curve represents a different defense against the attacks (GA: Gaussian attack; ZG: Zero-gradient attack; SF: Sign-flipping; and LF: Label-flipping).

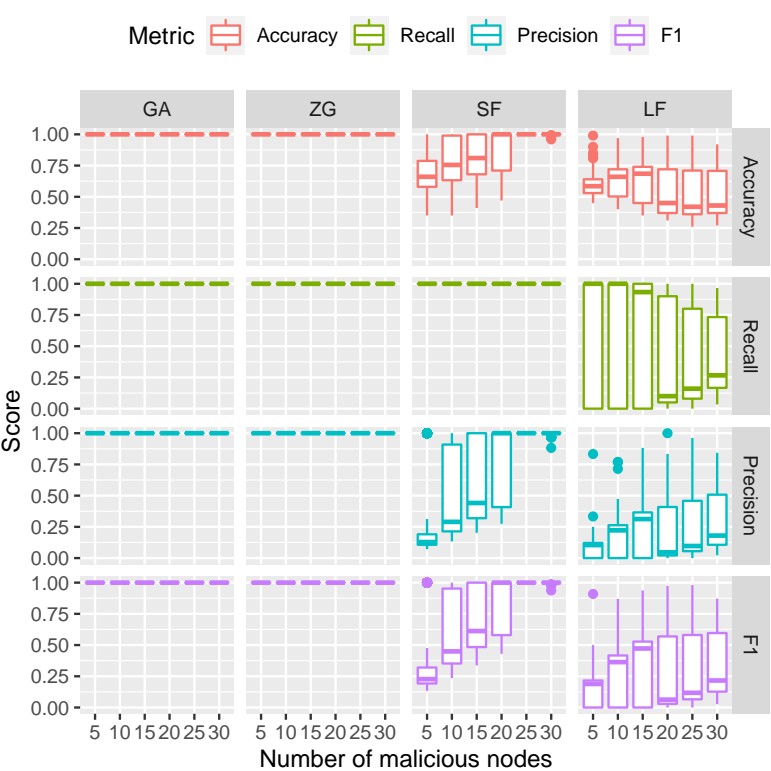

Figure 8: Classification performance of our proposed approach MANDERA (Algorithm 1) under four types of attack for MNIST data. GA: Gaussian attack; ZG: Zero-gradient attack; SF: Sign-flipping; and LF: Label-flipping. The boxplot bounds the 25th (Q1) and 75th (Q3) percentile, with the central line representing the 50th quantile (median). The end points of the whisker represent the Q1-1.5(Q3-Q1) and Q3+1.5(Q3-Q1) respectively.

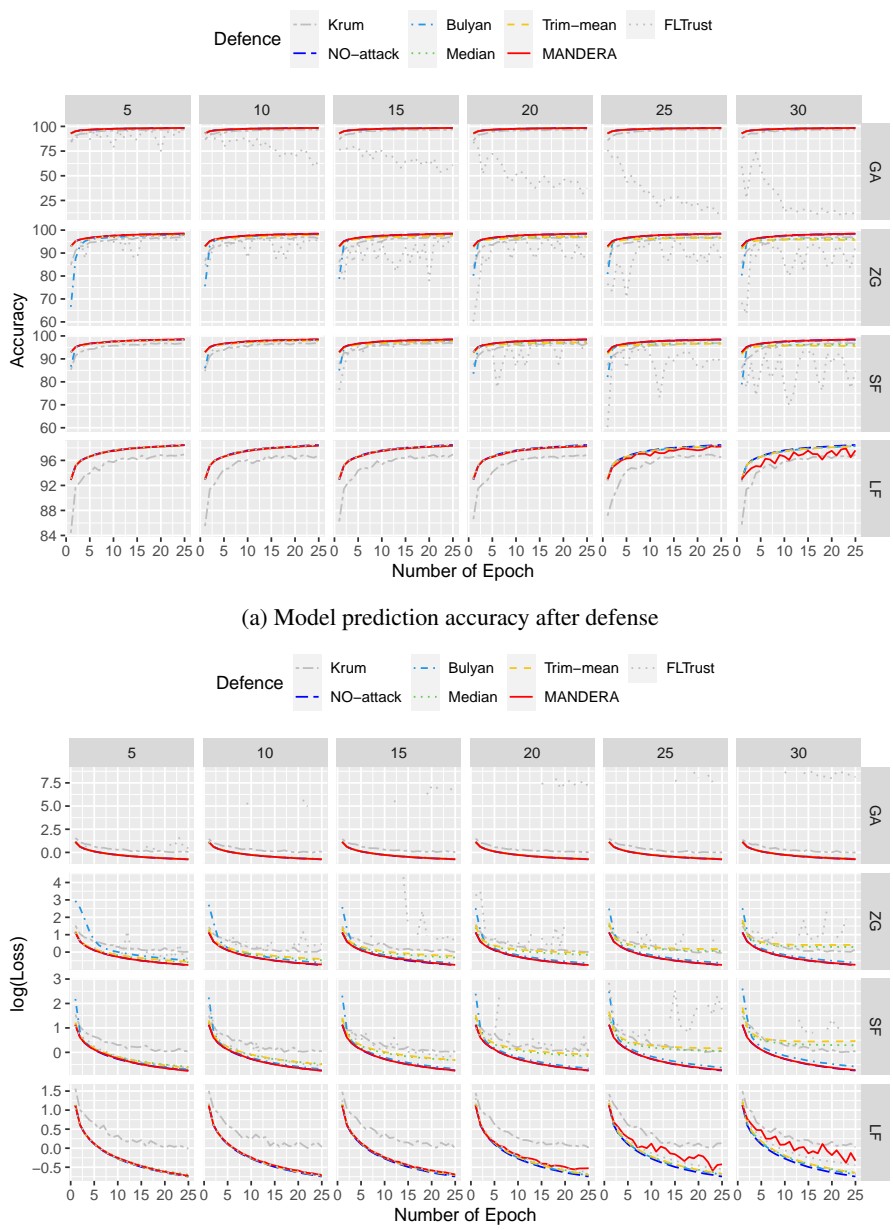

(a) Model prediction accuracy after defense

(b) Model prediction loss after defense

Figure 9: Model Accuracy and Loss for MNIST data at each epoch of training, each line of the curve represents a different defense against the attacks (GA: Gaussian attack; ZG: Zero-gradient attack; SF: Sign-flipping; and LF: Label-flipping).

