# OpenReview forum: "MANDERA: Malicious Node Detection in Federated Learning via Ranking"
_ICLR.cc/2022/Conference — ICLR 2022 Submitted_

### Official Review · Reviewer_FjLC · 2021-10-18

**Correctness:** 4
**Technical Novelty And Significance:** 2
**Empirical Novelty And Significance:** 2
**Recommendation:** 3
**Confidence:** 3

**Main Review:**

The paper proposes to use ranking instead of numeric values to cluster the user's update in federated learning to find malicious nodes. The authors list several attacks and provide analysis of the different behaviors between benign nodes and malicious nodes.

The work is rather complete. The results, AFAIK, are correct and the experiments are solid. The writing is clear and easy to follow. I really enjoy the illustrations in the paper especially Fig 3 and 4.

On the other hand, I have several concerns about the methodology itself. Most importantly, the paper does give any formal robustness guarantee. The method is designed based on several known attacks. Because security should not be preserved via obscurity, if this method is applied in real-world applications, the attackers can construct targeted attacks against this approach. For example, in high-dimensional case, the attackers can insert contaminated records close to the benign ones but still deviate the model from converging. To achieve real robustness, theoretically strong robustness [1] should be proved so as to prevent most available attacks (even not known). Second, the method is also not compatible with secure aggregation which prevents cumulative protection.

Overall, I do not think the paper is ready for publication until formal robustness guarantee is added.

[1] Wang, Lun, Qi Pang, Shuai Wang, and Dawn Song. "F2ED-Learning: Good Fences Make Good Neighbors."

**Summary Of The Paper:**

The paper proposes to transfer the update matrix to a ranking matrix before clustering to detect malicious nodes in federated learning.

**Summary Of The Review:**

I do not recommend acceptance because formal robustness guarantee is missing.

---

> ### Author Response · Authors · 2021-11-17
> **Responses to Reviewer FjLC**
>
> We appreciate the reviewer’s evaluation of the paper and the potential concerns raised. We have tried to clear up any misunderstandings within our work. To that end, below are the point-by-point replies to your questions and comments.
>
> 1) Regarding “the paper does give any formal robustness guarantee”. Thank you very much for proposing such an important point, on which we did not provide sufficient discussions previously. Following the reviewer’s suggestion, we have added a paragraph in section 2.6 to explicitly state the robustness guarantee offered by MANDERA. The reason why MANDERA can provide the formal robustness guarantee is given below. MANDERA can perfectly detect all the malicious nodes under mild conditions. So that following global update only uses updates from benign nodes. Such a strategy implies that a robustness guarantee can be achieved, because the MANDERA can exclude the malicious nodes, and thus proportion of bad data points in the aggregation step is 0.
> 2) Regarding “in high-dimensional case, the attackers can insert contaminated records close to the benign ones but still deviate the model from converging”. We have acknowledged the possibility of an attacker crafting a tailored example that maintains the ranking behaviour of malicious nodes to be like that of benign nodes. In such cases, MANDERA will fail to detect the malicious nodes. However, designing such an attack may require more information than just the updates of benign nodes. For example, a Gaussian attacker should pursuit ${\bar{s}}_m^2={\bar{s}}_b^2$ based on Theorem 1. In practice, it is not easy to implement attacks under the above constraint. The attacker needs to know the full distribution of the rank matrix to get ${\bar{s}}_b^2$, which are typically not realistic. Finally, MANDERA does not suffer from the curse of dimensionality compared to other techniques, such as Byzantine-robust mean estimators, instead, we have demonstrated that MANDERA would prefer the number of model parameters to be as large as possible to accommodate our theory (see Theorem 1, 2 and Corollary 3).
> 3) Regarding your comment “the method is also not compatible with secure aggregation which prevents cumulative protection”, many thanks for pointing out such an interesting problem, which we did not touch in our original manuscript. We totally agree with you that the current version of MANDERA is incompatible with secure aggregation, as we would lose the opportunity to get the rank matrix once the gradients are securely aggregated. However, it might be possible to upgrade the vanilla MANDERA to an advanced version that also supports secure aggregation under the framework of secure ranking [1,2]. As MANDERA only requires the rank matrix to work, the combination of MANDERA and secure ranking may provide a solution to the problem you pointed out. Due to the limited space in this paper, we cannot provide a detailed discussion about this issue.  In the revision, we have expanded the discussions in Section 4 to briefly talk about these ideas, leaving more details for future research.
>
> Reference
>
> [1] Zhang, Lan, et al. "Verifiable private multi-party computation: ranging and ranking." 2013 Proceedings IEEE INFOCOM. IEEE, 2013.
>
> [2] Lin, Hsiao-Ying, and Wen-Guey Tzeng. "An efficient solution to the millionaires’ problem based on homomorphic encryption." International Conference on Applied Cryptography and Network Security. Springer, Berlin, Heidelberg, 2005.

---

> > ### Comment · Reviewer_FjLC · 2021-11-21
> > **Response to rebuttal**
> >
> > I would like to thank the authors for the clarification.
> >
> > However, the biggest concern I have, that MANDERA is only proved to be robust to the several chosen attacks, is not solved. I would also like to mention that the chosen attacks in the paper are not very strong attacks. As a result, I seriously doubt the robustness guarantee given by the authors.
> >
> > The authors make strong claims about the robustness guarantee provided "MANDERA is guaranteed to detect all malicious nodes under typical Byzantine attacks with no prior knowledge or history about the participating nodes." "Theorem 1, 2 and Corollary 1 state that MANDERA can detect all the malicious nodes with probability 1". However,
> >
> > (1) I do not see why Thm 1 and 2 guarantee that all the malicious nodes will be detected. Actually the term "malicious nodes" is not well defined. If under Gaussian attack, the adversary happens to sample an update vector exactly the same as a normal vector, is the node still malicious?
> >
> > (2) The added robustness claim $||E[\hat{\mu}]-\mu||=0$ does not make sense because every unbiased estimator can satisfy it. A effective robustness guarantee should be something like $E[||\hat{\mu}-\mu||]\leq\delta$.
> >
> > (3) Even if Thm 1 and 2 indicates such robustness guarantee, the above claims should only hold under the chosen attacks but the authors do not mention the important conditions when making the claim.
> >
> > (4) Both Thm 1 and 2 are asymptotic results which does not directly guarantee the performance in practical use.
> >
> > Given these important concerns, I decide to lower my score accordingly. Please let me know if I misunderstand anything important. In brief, I would like to see a practical correct robustness guarantee that works under a clear assumption (such as those in Krum and Bulyan) instead of ad hoc guarantees against chosen attacks.

---

> > > ### Author Response · Authors · 2021-11-23
> > > **Second round response to reviewer FjLC:**
> > >
> > > We are deeply sorry that our previous response about the robustness guarantee has led to unexpected confusion and misunderstanding. In the updated manuscript, we have added a new theorem (Theorem 3) to precisely state the theoretical property of MANDERA on malicious node detection and aggregated message estimation for Byzantine attacks. Based on the theorem, for a particular FL problem with a finite number of nodes, as long as the sample size in each node (i.e., $N^*$) and the parameter dimensionality (i.e., $p$) are reasonably large, we would detect all malicious nodes and estimate the aggregated message $m_{b,:}$ consistently with probability one, under the three Byzantine attacks (i.e., Gaussian attack, sign flipping attack and zero gradient attack). Although these results are asymptotic, they do provide us clear guidance in practice and have been verified by a wide range of numerical experiments (see Fig 4-9 for the details).
> > >
> > > Our work is currently focused on the theoretical analysis of Byzantine attacks, which are comprised of Gaussian attacks, sign flipping and zero gradient attack. We believe that it is possible to prove similar results in a more general setting without concrete distributional assumptions on the malicious nodes. For example, under a much weaker assumption that the malicious nodes generate their message vectors from an unknown distribution $F^*$, which is different from the distribution $F$ for the benign nodes, we may still observe systematic behaviour differences between malicious nodes and benign nodes in the rank domain. Although the analytic calculation of the rank moments will become infeasible in such a setting, we may still be able to come up with a similar result based on the symmetry argument, since all benign nodes are exchangeable, and malicious nodes are exchangeable, but a benign node and a malicious node are not exchangeable.
> > >
> > > To establish such a theoretical framework for general attacks is beyond the scope of this paper, due to limited time and limited space. We do agree that working on a more general solution is an interesting direction for future work and will investigate in a future study.
> > >
> > > Our point-by-point responses to your comments in the new round are provided below. We hope such additional explanations could ease your concerns. We will greatly appreciate it if you would like to reconsider your decision based on the new inputs from us. Many thanks for your critical and deep thinking, and unique perspective, which has helped us understand the proposed method much better!
> > >
> > > 1) I do not see why Thm 1 and 2 guarantee that all the malicious nodes will be detected. Actually the term "malicious nodes" is not well defined. If under Gaussian attack, the adversary happens to sample an update vector exactly the same as a normal vector, is the node still malicious?
> > >
> > > **Response**: Your intuition is exactly right: if the adversary happens to sample an update vector exactly the same as a normal vector, we definitely cannot detect it as a malicious node. But, probability theory also tells us that the probability of such an event is 0 under the Gaussian attack. Thus, we do not really need to worry about such an ill and extreme case in practice.
> > >
> > > 2) The added robustness claim $[||E(\hat{\mu})-\mu||]=0$ does not make sense because every unbiased estimator can satisfy it. A effective robustness guarantee should be something like $E[||\hat{\mu}-\mu||]≤\delta$.
> > > **Response**: We apologize for the impropriety of notations in our previous response. We have removed the bad notation in the revision and provided a conclusion like $E[||\hat{\mu}-\mu||]≤\delta$ in the Theorem 3.
> > >
> > > 3) Even if Thm 1 and 2 indicates such robustness guarantee, the above claims should only hold under the chosen attacks but the authors do not mention the important conditions when making the claim.
> > > **Response**: We entirely agree with you that our claims only hold under the chosen attacks as you pointed out. We have highlighted the important conditions explicitly in Theorem 3 as you suggested.
> > >
> > > 4) Both Thm 1 and 2 are asymptotic results which does not directly guarantee the performance in practical use.
> > > **Response**: We totally agree with you that asymptotic results do have some limitations when applied to practical scenarios where we always have finite samples only. But, they still play a central role in theoretical analysis, and often provide useful guidance to us in many practical cases. Based on the experimental results conducted in this paper, we believe that we are in the luck case that these asymptotic results do work well in practice.

---

> > > > ### Comment · Reviewer_FjLC · 2021-11-23
> > > > **Response to follow-up rebuttal**
> > > >
> > > > Thanks for the reply. After reading it, I still believe that several important concerns about the validity of the robustness guarantee are not and cannot be addressed in the capacity of a rebuttal as they are far beyond the scope of the current paper like the authors said in the response. Without such realistic and generally robust guarantee, I am not sure whether the paper is ready for publication. I will discuss with the other reviewers and the AC and let you know if I have further questions.

---

### Official Review · Reviewer_4WMU · 2021-11-01

**Correctness:** 4
**Technical Novelty And Significance:** 3
**Empirical Novelty And Significance:** 2
**Recommendation:** 8
**Confidence:** 3

**Details Of Ethics Concerns:**

The paper is aware that it will likely spur an investigation into a new attack that might beat the proposed approach. However, even the results of beating the proposed approach will not cause any worse use of federated learning than is already used.

**Main Review:**

__Strong points__: The paper’s strengths are its novelty (both technical and theoretical), the nature of the problem it is attacking and its clarity.
- The paper presents a novel technique and way to address the Byzantine attack problem in federated learning (which is a significant problem) by identifying malicious nodes. The actual technique of using the moments of gradient ranks is especially intriguing because of its simple elegance and great theoretical guarantees. The paper does a great job showing how using the first and second moment of ranks can distinguish certain nodes under attack from a theoretical standpoint. The paper also does an excellent job going beyond theoretical guarantees, which hold with large numbers of samples, to show that the proposed MANDERA technique also works in practice and as good as other state-of-the-art methods
    - As more of a question, really, than a comment, but the assumption that the gradient ranks, $R_{:,j}$ , are statistically independent probably deserves some more questioning. I can appreciate that the paper shows that this is empirically so, with a few, limited examples, at the bottom of section 2.3, but is this always so at least for neural network models? And, if so, why?
- The paper is well written with only a few typos in it (.i.e “week” instead of “weak” in the last paragraph of section 2.3). The figures are clear and it is very easy for a reader to understand both why the technique works and how it works.
    - One area where the clarity could be improved is to provide a brief 1-2 sentence explanation of the label flipping attack in the first paragraph of section 3. All of the other techniques receive a much more robust explanation in the previous section with the theoretical grantees, so it would be good for the reader to better understand why that technique was used and what it is (at a high level).

__Weak points__: The paper is, overall, a strong paper with very few weak points. Mostly, the paper leaves one wondering about future work that could build on what is established within the paper, most of which the paper does comment on.
- For example, what about using more moments or combining rank statistics with other statistics of the gradient updates for more robust malicious node identification.
- What about the use of a better clustering technique? Looking at figure 3 and then looking at the recall problems with SF, it seems like a better clustering technique could probably solve this problem.
- As with other points, the paper does explicitly mention this in its ethics statement, but how would one design an attack to counter this defense?


**Summary Of The Paper:**

The paper presents a novel technique for defending against Byzantine types of attacks on federated learning systems.  The paper presents both theoretical justifications for why the using moments of ranks of gradient updates works as well as empirical justification showing that it works in practice.

**Summary Of The Review:**

I recommend accepting this paper, as it provides a novel technique with sound theoretical and convincing empirical justifications for attacking an important problem in the application of federated learning.

---

> ### Author Response · Authors · 2021-11-17
> **Responses to Reviewer 4WMU**
>
> Thanks for your detailed evaluations on our work. We greatly appreciate your constructive suggestions and positive comments. Below are our point-by-point replies to your questions and comments.
>
> 1) Regarding the independence assumption on the gradient ranks R:,j, as you pointed out, we are very grateful to give us an opportunity to further clarify this important issue. We totally agree that it is extremely rare to observe truly independent gradient ranks in practice. In other words, the independence assumption is always wrong in practice. However, according to the famous saying “all models are wrong, but some are useful” by George Box, one of the greatest statisticians in the last century, we believe that the independence assumption we proposed in this study is also a “wrong but useful model”. As we have demonstrated in the paper via numerical experiments, the involved gradient ranks empirically behave independently, although they are surely dependent on each other. The extremely weak dependence among these ranks makes it reasonable to approximate the complicated reality with a simple independent model, which can provide us the theoretical guarantee that could well predict the behaviour of the system in practice with small errors. On the other hand, such an independence assumption could be relaxed to a weaker “uncorrelated assumption”, which assumes that ranks are uncorrelated with each other. Adopting the weaker assumption will result in a change of convergence type of our theorems from the “almost surely convergence” to “convergence in probability”, but with no essential influence to the MENDERA algorithm itself. We have added a few sentences in the revised paper to clarify these issues.
> 2) Regarding the typos, we have proofread the paper for multiple rounds and corrected all typos and grammatical errors we have found.
> 3) Regarding your suggestion to provide a brief description of the Label Flipping attack, we have taken onboard your advice in providing a high-level description of the Label Flipping attack to contextualize its presence in the revised paper, which sits in the first paragraph of section 3.
> 4) Regarding your suggestion on “using more moments or combining rank statistics with other statistics of the gradient updates for more robust malicious node identification”, we have implemented additional experiments to test whether involving higher-order moments could lead to detection improvement. Our initial tests indicated that there was no clear improvement compared to our current algorithm, we believe this will warrant the need for further investigation and derivation in future work.
> The following two facts could help explain such a phenomenon. First, in all experiments involved, the first two moments have provided enough information for malicious node detection, leaving little room for higher-order moments to improve performance. Second, the higher-order moments are numerically unstable and sensitive to outliers and noise in the data, and thus suffer from a low signal-noise ratio.
> In the event that the first two moments are not sufficient to separate the malicious nodes from the benign ones, involving higher-order moments may play an important role under the framework of MENDERA. But unfortunately, we did not find such an example in this study.  We hope these efforts can provide useful insights to your comments and help stimulate continued interest in the area.
> 5) To provide an answer to your question “What about the use of a better clustering technique?”, we have performed an evaluation to replace the K-means algorithm in the current setting with other clustering methods, including hierarchical clustering and Gaussian mixture models. We found that performance of MANDERA is quite robust to different choices of the clustering methods. We have included these additional comparisons in Appendix E.
> 6) Regarding your question on “how would one design an attack to counter this defense”, we think the most straightforward way is to manipulate the malicious nodes so that they can generate similar moments of gradient ranks as the benign ones. For example, a Gaussian attacker should pursuit ${\bar{s}}_m^2={\bar{s}}_b^2$ based on Theorem 1. In practice, however, it is not easy to implement attacks under the above constraint. First, the attacker needs to know the full distribution of the rank matrix to get ${\bar{s}}_b^2$, which is typically not realistic. Second, such a constraint leaves little room for the attacker to manipulate the malicious nodes, making it much more difficult to design effective attacks.

---

> > ### Comment · Reviewer_4WMU · 2021-11-18
> > **Thank you for the replies**
> >
> > Dear Authors,
> >
> > Thank you for the replies to the questions I posed. I very much appreciate you adding the extra details (i.e. use of other clustering techniques, more explanation on independence assumption) that you did, as I believe it makes a more complete paper. I have also briefly looked through some of the other reviewer comments - in particular, the comment about robustness guarantees which was an important shortcoming - and, at a cursory look, believe you have addressed those as well. As such, I intend to keep my recommendation on the paper. And, I have no further comments or suggestions at this time.

---

### Official Review · Reviewer_5jZy · 2021-11-03

**Correctness:** 4
**Technical Novelty And Significance:** 3
**Empirical Novelty And Significance:** 3
**Recommendation:** 6
**Confidence:** 3

**Main Review:**

(+) The main strength of the paper lie in the novel introduction of using the rank domain in order to detect malicious nodes. All of the claims are properly supported and experiments and results are clear.

(-) The weaknesses in the paper are in the number of datasets/attacks. Having more experiments on a larger range of datasets and attacks will support claims more. Additionally, the inclusion of a strong, recent Byzantine defense algorithm in the robust learning area, such as FLTrust, will also help show the performance against state-of-the-art robust learning defense algorithms.


**Summary Of The Paper:**

The paper focuses on Byzantine defense through malicious node detection in a Federated Learning setting. Namely, by ranking the gradients and then computing the mean/SD, the paper shows that the malicious and benign clients will cluster separately. Assuming that the number of malicious clients is fewer than the number of benign clients, and the clusters correctly separate the malicious from benign, the smaller cluster is removed, and training is done on the gradients in the larger cluster. Appropriate experiments are done to show the ability of the model in malicious node detection along with analysis of performance and computational requirements.

**Summary Of The Review:**

This paper introduces a new perspective in Byzantine defense in comparison to typical robust learning defense or other detection defenses. While the algorithm and experiments done are relatively simple, the paper serves more as a starting point for research and application in the new domain. Claims are properly supported and the method's competitive performance compared to other methods is promising.

---

> ### Author Response · Authors · 2021-11-17
> **Responses to Reviewer 5jZy**
>
> Thank you for your receptive outlook on our work! We have considered your comments carefully. Below are our point-by-point responses to your questions and comments.
> 1)	Regarding “Having more experiments on a larger range of datasets and attacks will support claims more”, we have provided an additional evaluation of the MNIST handwritten digit dataset in Appendix G. On this dataset, we repeat both our extensive evaluation of MANDERA, and the comparative evaluation between MANDERA and other defence methods. We find out that existing findings continue to hold for the MNIST dataset.
> 2)	Regarding “the inclusion of a strong, recent Byzantine defence algorithm in the robust learning area, such as FLTrust, will also help show the performance against state-of-the-art robust learning defence algorithms”, we have taken on your recommendation in expanding our experimentation by including FLTrust into the list of competing defence methods of MENDERA. Performance of FLTrust on different datasets, including FASHION-MNIST, CIFAR-10 and MNIST, are reported in Section 3 and Appendix G, suggesting that the MENDERA is still a competitive defence method with unique advantages.

---

> > ### Comment · Reviewer_5jZy · 2021-11-17
> > **Missing Appendix G**
> >
> > I appreciate the additional experimental details.
> >
> > "Performance of FLTrust on different datasets, including FASHION-MNIST, CIFAR-10 and MNIST, are reported in Section 3 and Appendix G, suggesting that the MENDERA is still a competitive defence method with unique advantages."
> >
> > I do not see an Appendix G --- the last one in the paper is Appendix E.

---

> > > ### Author Response · Authors · 2021-11-17
> > > **Revision uploaded**
> > >
> > > The revised paper was just uploaded. Please check.

---

> > > > ### Comment · Reviewer_5jZy · 2021-11-25
> > > > **Good additional experimental results, but two clarification questions**
> > > >
> > > > It is good to see the additional experimental results for CIFAR-10 and Fashion MNIST (Figure 7(a) and (b)). It is clear from this result that MANDERA outperforms FLTrust (the baseline I was most interested in).
> > > >
> > > > Two clarification questions:
> > > > 1) What configuration of FLTrust was used? The size of the validation set used in FLTrust can make a difference as can the degree of non-iid bias.
> > > >
> > > > 2) What does the quantity on the columns (5, 10, ..., 30) represent? Please clarify that. The Appendix G does not describe the experiment used to create Figure 7. That would have been useful.
> > > >
> > > > Overall I remain positive on the paper.

---

> > > > > ### Author Response · Authors · 2021-11-25
> > > > > **Further Responses to Reviewer 5jZy**
> > > > >
> > > > > Thank you for your positive comments. We are appreciated if you could update your score to reflect your positive view on our paper. Below are the further clarifications. We would be happy to add these clarifications in the camera ready version.
> > > > >
> > > > > 1). What configuration of FLTrust was used? The size of the validation set used in FLTrust can make a difference as can the degree of non-iid bias.
> > > > >
> > > > > **Response**: we understand your description of ‘the validation set’ for FLTrust as the ‘root dataset’. In our paper, the root dataset is chosen as follows: the 60000 samples are randomly divided into 100 chunks, and 10 chunks are randomly chosen as the root dataset. Therefore, the resulting sample size in the root dataset is 6000. While the default sample size in FLTrust is 100. Our setting should be beneficial to FLTrust. As you can see, the root dataset is a set of iid samples. More details about data split can be found in ref [1].
> > > > >
> > > > > 2). What does the quantity on the columns (5, 10, ..., 30) represent? Please clarify that. The Appendix G does not describe the experiment used to create Figure 7. That would have been useful.
> > > > >
> > > > > **Response**: We apology for the confusion. The number on each column represents the number of malicious nodes. The relevant appendix section is now H. More descriptions will be added.
> > > > >
> > > > > **Reference**
> > > > >
> > > > > [1] Tolpegin, Vale, et al. "Data poisoning attacks against federated learning systems. " European Symposium on Research in Computer Security. Springer, Cham, 2020.

---

### Decision · Program_Chairs · 2022-01-20

**Decision:**

Reject

**Comment:**

This manuscript proposes a ranking approach to identify Byzantine agents in federated learning. Distinct from existing methods, the mitigation is implemented by computing ranks for each gradient, then computing rank statistics across agents. The primary intuition is that adversarial agents can be identified by examining these rank statistics.

There are three reviewers, all of whom agree that the method addresses an interesting and timely issue -- giving the growing interest in both Byzantine-robust learning and federated learning in the community. However, reviewers are mixed on the paper score -- with a strong accept a weak accept, and a strong reject. Common issues raised include the generality of the approach beyond the outlined attacks,
Other issues brought up, but addressed in the rebuttal include some weaknesses in the evaluation and comparison to additional baselines. There is also an interesting discussion of using higher-order statistics, which does not seem to help the methods when evaluated by the authors. Nevertheless, after reviews and discussion, the reviewers are mixed at the end of the discussion.

The area chair finds, first, that the paper is much improved, and much more applicable in the updated form than in the original version. However, the area chair agrees with the reviewer who notes that the moniker "Byzantine-robust" implies the methods should be provably robust to worst-case adversaries, not only to a selected set of adversaries with pre-selected attacks. The specified setting may be too narrow for interest by the community. To this end,  the area chair suspects that the method may be robust to a more general set of attacks than noted -- working to outline sufficient conditions for robustness would significantly strengthen this work. The asymptotic nature of the robustness guarantees is also of concern.

An additional concern of the area chair is that the system setting investigated assumes gradient communication and IID data across devices. While this is not an issue on its own, the setting is closer to distributed learning than federated learning, where one generally communicates model updates, or model differences after multiple local updates, and not gradients. This difference can have a significant effect on robustness methods that depend on identifying benign vs. adversarial statistics of parameters. Non-IID data is also common in the federated setting, though this is less concerning, as robust methods for non-IID settings are only now emerging. A simple fix for this issue would be to rename the setting from "Federated" to "Distributed."

Authors are encouraged to address the highlighted technical concerns in any future submission of this work. The primary concern may simply be a naming issue (i.e., removing "Byzantine" might fix this concern. Nevertheless, taken together, the opinion of the area chair is that the manuscript is not ready for publication. Again, the area chair believes that many of the issues noted can be fixed, the paper can be strengthened, and this paper may be publishable with limited additional work.